# Cell-Free Circulating (Tumor) DNA before Surgery as a Prognostic Factor in Non-Metastatic Colorectal Cancer: A Systematic Review

**DOI:** 10.3390/cancers14092218

**Published:** 2022-04-29

**Authors:** Suzanna J. Schraa, Karlijn L. van Rooijen, Miriam Koopman, Geraldine R. Vink, Remond J. A. Fijneman

**Affiliations:** 1Department of Medical Oncology, University Medical Center Utrecht, Utrecht University, Heidelberglaan 100, 3584 CX Utrecht, The Netherlands; s.j.schraa@umcutrecht.nl (S.J.S.); kl.van.rooijen@stjansdal.com (K.L.v.R.); m.koopman-6@umcutrecht.nl (M.K.); g.vink@iknl.nl (G.R.V.); 2Department of Research and Development, Netherlands Comprehensive Cancer Organisation, Godebaldkwartier 419, 3511 DT Utrecht, The Netherlands; 3Department of Pathology, Netherlands Cancer Institute, Plesmanlaan 121, 1066 CX Amsterdam, The Netherlands

**Keywords:** circulating tumor DNA, circulating cell-free DNA, liquid biopsy, DNA methylation, colorectal cancer, surgery, prognosis, survival, systematic review

## Abstract

**Simple Summary:**

Patients with colorectal cancer without distant metastases are often cured by surgical tumor resection. Follow-up is needed because of the risk of disease recurrence. Patients at risk of disease recurrence may benefit from additional chemotherapy. Detection of cell-free circulating tumor DNA (ctDNA) after surgery reflects the presence of remaining cancer cells and is associated with a very high risk of recurrence. Therefore, postsurgery detection of ctDNA is a promising approach to accurately identifying high-risk patients. However, postsurgery ctDNA analysis is challenging. Moreover, in some patients, chemotherapy before surgery might be more beneficial than chemotherapy after surgery. In this review, we provide an overview of current knowledge regarding the association between ctDNA detection before surgery and the risk of recurrence and conclude that the current literature is insufficient to determine this association. Dedicated studies that primarily focus on ctDNA before surgery in colorectal cancer patients are needed.

**Abstract:**

Identification of non-metastatic colorectal cancer (CRC) patients with a high risk of recurrence after tumor resection is important to select patients who might benefit from adjuvant treatment. Cell-free DNA (cfDNA) and circulating tumor DNA (ctDNA) analyses after surgery are promising biomarkers to predict recurrence in these patients. However, these analyses face several challenges and do not allow guidance of neoadjuvant treatment, which might become a novel standard option in colon cancer treatment. The prognostic value of cfDNA/ctDNA before surgery is unclear. This systematic review aims to provide an overview of publications in which the prognostic value of presurgery cfDNA/ctDNA in non-metastatic CRC patients was studied and is performed according to PRISMA guidelines. A total of 29 out of 1233 articles were included and categorized into three groups that reflect the type of approach: measurement of cfDNA, ctDNA somatic alterations, and ctDNA methylation. Overall, a clear association between presurgery cfDNA/ctDNA and the outcome was not observed, but large studies that primarily focus on the prognostic value of presurgery cfDNA/ctDNA are lacking. Designing and performing studies that focus on the value of presurgery cfDNA/ctDNA is needed, in addition to standardization in the reporting of cfDNA/ctDNA results according to existing guidelines to improve comparability and interpretation among studies.

## 1. Introduction

Surgical resection remains the cornerstone of treatment in non-metastatic colorectal cancer (CRC) patients. Optimal reduction in recurrence risk with (neo)adjuvant treatment requires a more personalized approach. Approximately 20% of stage I–III CRC patients treated with a surgical resection will eventually develop distant metastasis, which has a poor prognosis [1,2,3]. Adjuvant and neoadjuvant therapies are offered based on tumor node metastasis (TNM) staging and clinical parameters, which reduces the chance of disease recurrence. However, significant over- and undertreatment exists as the accuracy to predict recurrence based on these parameters is unsatisfactory. A more accurate way to identify CRC patients with a high risk of recurrence remains an unmet clinical need.

Adjuvant chemotherapy is offered to patients with stage III and a subgroup of high-risk stage II colon cancer patients. Neoadjuvant therapy, consisting of radiotherapy and/or chemotherapy, is mostly reserved for patients with rectal cancer and was proven to be more effective than adjuvant therapy by reducing the risk of incomplete resection and tumor cell shedding during surgery [4,5,6]. Neoadjuvant therapy is currently not recommended in guidelines for colon cancer. However, the promising results of the FOxTROT trial and the recently published meta-analysis on this subject may shift treatment from adjuvant toward neoadjuvant in the near future for locally advanced (cT3-T4) colon cancer, which may improve survival outcomes [7,8].

Presurgery selection of patients that are prone to disease recurrence will allow clinicians to tailor (neo)adjuvant therapies, which may benefit only a select group of patients and carry risks of significant toxicities. This warrants a new prognostic biomarker, which ideally also helps to select the right patient for adjuvant treatment.

With a neoadjuvant approach, reliable TNM staging is challenging as clinical and radiological TNM staging are often inaccurate. A potential solution could be found in liquid biopsies, an emerging and rich source of biomarkers that is being studied in multiple cancer types using different approaches. Liquid biopsies encompass a wide range of technologies to derive tumor information from body fluids and include analysis of circulating tumor cells (CTCs), circulating cell-free DNA (cfDNA), circulating tumor DNA (ctDNA), and microRNAs (Figure 1). ctDNA is the fraction of cfDNA that is released by tumor cells [9]. ctDNA provides a comprehensive view of the tumor genome, has a short half-life allowing real-time monitoring, and is considered one of the most promising noninvasive biomarkers to be implemented in daily cancer care [10].

In non-metastatic CRC, ctDNA is being studied as a biomarker for the prediction of recurrence. Multiple studies show that the presence of ctDNA after surgery indicates minimal residual disease (MRD), resulting in recurrence rates of around 80–100% [11,12,13].

However, ctDNA analysis postsurgery faces several challenges. Detection of ctDNA in the MRD setting is difficult as only very small amounts of ctDNA are present in the blood. Technically this requires complex, tissue-informed analysis with ultradeep sequencing. So far, ctDNA can be detected in only 5–8% of stage II patients, while recurrence rates are approximately twice as high [11]. Moreover, the optimal timing of postsurgery blood sampling for ctDNA analysis is unclear. Recent studies suggest that blood withdrawals within 4 weeks of surgery have lower sensitivity as normal cfDNA levels rise due to surgery-induced tissue injury [14]. However, for a ctDNA-guided approach to offer adjuvant chemotherapy, the turnaround time of the ctDNA test result should be available within 8 weeks after surgery to guide clinical decision making [15].

Analysis of cfDNA/ctDNA before surgery could overcome these challenges. Higher levels of presurgery compared to postsurgery ctDNA increase the detection rate, and blood withdrawal before resection creates a larger time window for ctDNA analysis. Therefore, adjuvant chemotherapy could be started early after surgery, and also neoadjuvant treatment could be considered. However, the prognostic value of cfDNA and ctDNA presurgery is unclear. Currently, most studies that analyze cfDNA or ctDNA in treatment-naïve patients are aimed at the detection of postsurgery minimal residual disease, and little is known about the prognostic value of presurgery cfDNA/ctDNA [16].

The objective of this systematic review is to provide an overview of publications in which the prognostic value of cfDNA and ctDNA before surgical resection of the primary tumor in stage I–III CRC patients is studied and to evaluate the association between presurgery cfDNA/ctDNA and outcomes.

## 2. Materials and Methods

### 2.1. Study Design

A systematic review was conducted to provide an overview of available evidence regarding the analysis of cell-free circulating (tumor) DNA before surgery to predict outcomes in non-metastatic CRC. The systematic review was performed according to the Preferred Reporting Items for Systematic Reviews and Meta-analysis (PRISMA) protocol [17]. Our protocol was registered at Open Science Framework (registration available at: https://osf.io/ujsqp, accessed on 11 April 2022) [18].

### 2.2. Search Strategy

On 20 April 2021, a search was performed using PubMed and Embase. The following search query was used: (“colon cancer*” OR “colorectal cancer*” OR “colon carcinoma*” OR “colo rectal cancer*” OR “rectal carcinoma*” OR “rectal cancer*” OR CRC* OR “bowel cancer*” OR “Colonic Neoplasms” OR “colonic cancer*” OR “colonic neoplasm*” OR “colon neoplasm*” OR “colorectal adenoca*”) AND (“circulating tumor DNA” OR ctDNA OR “cell free DNA” OR “liquid biops*” OR “circulating tumour dna” OR “circulating free dna” OR “serum DNA”). Results were uploaded in Endnote, which was used to remove duplicates. The remaining results were uploaded in Rayyan to be independently reviewed by two researchers (SJS, KLR). In case of discrepancies between the reviewers, a third arbiter (RF) decided whether or not to include the article concerned. Titles and abstracts were screened, followed by a full-text revision to assess our selection criteria.

### 2.3. Inclusion and Exclusion Criteria

Included studies, both retrospectively and prospectively performed, were those reporting on the detection of presurgery cfDNA/ctDNA in non-metastatic CRC with a clinical outcome of recurrence or survival. Studies that included fewer than 10 patients and studies reporting on neoadjuvant-treated patients were excluded. In addition, studies focusing on circulating tumor cells (CTC), extracellular vesicles, and RNA rather than ctDNA or cfDNA were excluded. Review articles, studies in animal models or cell lines only, and studies that were not published in English were excluded.

### 2.4. Evidence Synthesis

The study publications were reviewed by abstracting the following variables into a database: name of the first author, year of publication, study design, inclusion criteria, timing of blood collection, biomarkers and assays that were used, number of patients, tumor type and stage, patient characteristics, sensitivity of biomarker, recurrence data, and survival data. Publications within a certain category were compared. Because many different assays and outcomes are being reported in cfDNA/ctDNA studies, no meta-analysis was performed.

### 2.5. Quality of Evidence

The risk of bias (RoB) concerning our research question was assessed using the Quality in Prognosis Studies (QUIPS) tool [19]. Quality ratings were assigned by the first author of this review. Moderate and high scores were confirmed by the second author. An overall score was defined for each article having high, moderate, or low RoB using the following criteria: if all domains were classified as having low RoB or up to one moderate RoB, then this article was classified as low RoB. If one or more domains were scored as having high RoB, or ≥ 3 moderate RoB, then this paper was classified as high RoB. All papers in between were classified as having moderate RoB. All studies in which stage IV patients were included, and in which these patients were not excluded from outcome analysis nor performed a multivariate analysis, scored a high RoB in the study confounding domain and therefore also had an overall high RoB score. All studies were covered in this review regardless of their RoB score.

## 3. Results

In total, 1233 publications were screened on title and abstract (Figure 2). A total of 132 publications were selected for full-text analysis, of which 29 met the inclusion criteria and were therefore further analyzed. We divided the 29 included articles into 3 different categories based on their approach: quantification of total cfDNA, detection of ctDNA somatic alterations, and detection of ctDNA epigenetic alterations by methylation assays (Figure 1). This way, 4 studies were categorized as cfDNA studies (14%), 12 studies focused on somatic alterations (41%), and 13 studies studied methylation (45%) (Table 1). A complete overview of study characteristics can be found in Appendix A. Of the included articles, 45% were scored as having high RoB, mostly due to the inclusion of stage IV patients in the outcome analysis (Appendix A).

### 3.1. Cell-Free DNA

cfDNA concerns the total amount of DNA fragments in the blood. These fragments are present in both healthy and diseased subjects. In cancer patients, cfDNA contains both tumor-specific and non-tumor-derived DNA. Background cfDNA typically has a length of 166 base pairs (bp), while tumor-specific ctDNA is, on average, shorter [48,49]. Biological processes with increased cell turnover lead to a greater amount of cfDNA, so increased levels are seen in the context of inflammation, trauma, surgery, and in cancer. Previous studies indeed showed higher levels of cfDNA in cancer patients than in healthy individuals [50,51]. In the metastatic setting of CRC, cfDNA is a known independent prognostic factor for disease-free and overall survival [52]. However, in early-stage CRC, the prognostic value of cfDNA/ctDNA is not yet established.

Measurement of cfDNA levels is a quantitative approach but non-specific for malignancy. The levels of postoperative cfDNA can be affected by the extent of surgery and postoperative complications. Therefore, if a prognostic value could be established in the preoperative setting, it would be of great interest.

#### Summary of Included Studies

Four publications focusing on cfDNA quantification were included in this review (Table 1). Two studies found a significant association between cfDNA levels and disease-free survival, but stage IV patients were included in the analysis [20,23].

Guadalajara et al. did not find a correlation between the total amount of cfDNA and disease-free survival (DFS) or overall survival (OS) [21]. The follow-up time was limited (mean 18 months), but a trend toward a worse prognosis was seen for patients with cfDNA concentrations above 60 ng/μL.

A prospective study from Israel included fewer patients, but a follow-up of 5 years was reached [20]. In a multivariate Cox regression analysis, they found a significant effect of high cfDNA (>800 ng/mL) on both DFS (HR 6.03) and OS (HR 8.55). In a more recent study, cfDNA dynamics were analyzed perioperatively, and samples were collected before surgery and at several time points up to 5 days after surgery [22]. Twenty patients were included and were followed for at least two years. The preoperative level of cfDNA in three patients with a recurrence was slightly higher than in patients without a recurrence, but no statistical analysis was performed.

In a larger Chinese population, cfDNA concentration was significantly and independently correlated with disease and progression-free survival [23]. Although not reported in detail, a minimum of 42% of the patients had metastatic disease at enrollment, making interpretation for non-metastatic CRC patients difficult.

### 3.2. Somatic Alterations

In cancer patients, tumor cells release DNA fragments in the blood as a result of apoptosis, necrosis, and secretion [53]. ctDNA is the subset of total cfDNA that contains tumor-specific somatic alterations, such as single nucleotide variants (SNV) and structural variants.

Most common mutations that have been identified in the primary tumor of CRC patients include mutations in APC, TP53, KRAS, and BRAF [54]. These mutations can be related to prognosis and outcomes in certain cases. For instance, patients with a BRAF V600E and RAS wildtype tumor seem to have a favorable prognosis compared to patients with a mutation in these genes, independently of mismatch repair status [55,56,57].

Detection of CRC-related mutations in blood to confirm the presence of ctDNA is challenging in early-stage CRC, especially in the postsurgery setting when tumor load and, therefore, mutation allele frequency (MAF) is either absent or very low. Moreover, non-tumor-derived genetic alterations, such as those related to clonal hematopoiesis, could cause false-positive results. Therefore, a tumor tissue-guided approach is often needed to reach sufficient sensitivity and specificity. Analysis of plasma instead of serum is preferable because of the dilution of ctDNA by non-tumor DNA in serum [56].

#### Summary of Included Studies

In 11 publications, the prognostic value of ctDNA by detecting somatic alterations was studied (Table 1). In five studies, a significant association was found, in four articles, a non-significant trend was described, and in three studies, no significant association could be found. Several studies analyzed mutations in a single gene. One study included 160 patients with a known KRAS mutation in tumor tissue [26]. In 9 out of 54 (17%) non-metastatic CRC patients, the KRAS mutations were detectable in plasma using a combination of mass spectrometry and an ultrahigh-sensitivity PCR-based assay. Of patients with detectable ctDNA, 89% developed a recurrence, compared to 78% of patients without detectable plasma KRAS mutations, which was not statistically significant. However, the high recurrence rate of 80% for the overall group of non-metastatic CRC patients suggests a selection bias and raises questions about interpretation. A Japanese study used a digital PCR (dPCR) assay to detect KRAS mutations in the plasma of 180 CRC patients without knowledge of tissue mutation status [31]. Patients with detectable KRAS mutations had a higher recurrence rate (27% vs. 9%) and an inferior recurrence-free survival (RFS; HR 2.18). However, as the mutation status of tumor tissue is unknown, this study cannot distinguish between the detection of ctDNA as a prognostic factor or the known association between KRAS mutated tumors and prognosis.

Because only 40–45% of CRCs harbor a KRAS mutation and, therefore, could potentially be detected in blood, Lecomte and colleagues combined detection of KRAS mutations with p16 methylation using PCR techniques [25]. This way, in 25/45 (56%) of non-metastatic CRC patients, the tumor harbored one or both of these genetic alterations. Their blood samples could be analyzed for the presence of ctDNA. Patients with detectable ctDNA (68%) had significant worse RFS (2-year RFS: 66% vs. 100%, *p* = 0.044) and OS. Another study group used single-strand conformation polymorphism (SSCP)-PCR to detect either KRAS, APC, or TP53 mutations in both tissue and serum [24]. Using this PCR-SSCP, large PCR products are amplified, sometimes even larger (>300 bp) than an average ctDNA fragment. This warrants the question of what mutations are being detected using SSCP-PCR for serum analyses. Are mutations detected in ctDNA fragments or, for example, in circulating tumor cells? Nevertheless, the presence of at least one mutation in serum was observed in 36/78 (46%) of patients with tumors containing at least one of these mutations and was associated with a higher recurrence rate (75% vs. 10%, *p* < 0.001). Detection of ctDNA was also associated with lymph node metastases and, therefore, a more advanced stage. No multivariate analysis was performed.

In six studies, multiple genes were investigated to identify mutations in plasma. The highest number of non-metastatic colon cancer patients was included in a retrospective cohort [29]. dPCR was used to evaluate the presence of 25 common mutations in KRAS, BRAF, and NRAS in the presurgery blood. Patients with detectable RAS mutations in both tumor and serum had worse DFS (HR 2.18) compared to patients with a RAS mutation in the tumor only. Detectable BRAF mutations in serum were only correlated with worse DFS in patients with proficient mismatch repair (pMMR) tumors.

Combining dPCR and NGS could increase sensitivity in detecting ctDNA before surgery, according to an Italian study [32]. However, follow-up was available for 10 patients only. Three patients relapsed, all having detectable ctDNA before surgery. Of non-relapsing patients 4/7 (57%) had detectable ctDNA. In a publication from 2017, whole-exome sequencing (WES) was performed on the tissue to design a personalized dPCR assay for detecting both chromosomal rearrangement structural variants and single nucleotide variants (SNVs) [28]. Patients with recurrent disease more frequently had presurgery detectable ctDNA than patients who remained free of disease (80% vs. 55%). In addition, in this study, the numbers of included patients were relatively small. More patients were included in a more recent study of the same Danish research group, using a similar approach but focusing exclusively on SNVs [12]. No significant association between presurgery ctDNA and outcome was observed. A Spanish study performed targeted NGS on tumor tissue to design a personalized dPCR assay for ctDNA detection [30]. After an adequate follow-up period, 18 recurrences were detected in 94 patients, but no relationship was found between baseline ctDNA detection and DFS (HR 0.33–2.69).

In a retrospective study, including 27 non-metastatic CRC patients, a significant difference was observed in both DFS and OS for patients with and without detectable ctDNA [33]. Interestingly, ctDNA was analyzed both as a binary (detectable or undetectable) and as a continuous variable (using mutant allele fractions). A comparable NGS approach was used in a larger cohort with a shorter follow-up time [34]. Only patients with detectable ctDNA were included in the follow-up analysis. No further analysis was performed, but a MAF heat plot showed no discriminative value between patients with high or low risk of recurrence.

In a Danish study, only 11 patients were retrospectively included [27]. Five out of six patients with a recurrence had detectable ctDNA based on detection of structural variants, compared to three out of five patients without a recurrence.

### 3.3. Methylation

Tumor DNA not only differs from normal DNA by the presence of mutations. Epigenetic modifications are even more frequent than somatic alterations in cancer development [58]. A major feature is the hypermethylation of CpG islands in gene promoter regions, causing altered expression of these genes. This hypermethylation occurs early in cancer development and therefore is studied mostly for early detection purposes. Hypermethylation of the SEPTIN9 gene (SEPT9) is marked as one of the most evaluated biomarkers in CRC. In 2016, the Epi ProColon biomarker assay detecting SEPT9 methylation received FDA approval as a CRC screening test. No tissue is needed for methylation assays to reach an accurate specificity, which is an important advantage in the presurgery setting.

#### Summary of Included Studies

Thirteen studies analyzed the prognostic value of ctDNA detection by methylation assay in CRC (Table 1). Three studies focused on SEPT9 specifically. Song et al. detected SEPT9 hypermethylation in blood in 87% of patients [43]. For OS analysis, 82 patients were available, showing decreased OS for SEPT9-positive patients (HR 2.53, 95% CI 1.60–3.90) in univariate analysis. In a study with only 10 patients, a similar detection rate was found [45]. All four patients with a recurrence were SEPT9-positive before surgery, but three out of these four patients had (limited) metastatic disease at inclusion. Disease-specific mortality increased in patients with SEPT9, RARβ2, and SOX17 hypermethylation, according to Constâncio and colleagues [44]. However, the detection rate was low (8% for SEPT9), and the number of events was limited (9 cases) due to limited follow-up time. Jin et al. detected SEPT9 hypermethylation in 89% of patients [46]. No correlation was found between ctDNA status or level and recurrence in 82 patients with 24 recurrences. Interestingly, the methylation assay was compared to targeted NGS for somatic mutations in six patients with a relapse; five of them had detectable ctDNA. The assays showed concordant results in the preoperative samples.

The prognostic value of SEPT9 is therefore uncertain, but more CpG regions are being studied. For example, hypermethylation of HTLF is associated with an increased recurrence risk (HR 2.5, 95% CI 1.1–5.6) and worse RFS, according to Herbst et al. [40]. In contrast with the SEPT9 studies, HLTF hypermethylation was detected in only 12% of patients. In a previous study that was published in 2006, this study group found an increased risk of death (RR = 3.4, 95% CI 1.4–8.1) for HLTF hypermethylation [39]. The prognostic value increased when also taking HPP1 hypermethylation into account, but this was not confirmed in their most recent publication. Studies focusing on hypermethylation of SFPR1, OSMR, TWIST1, FLI1, and AGBL4 were not able to confirm an association with recurrence or survival [38,42]. Matthaios et al. studied methylation of APC and RASSF1A, which could be detected in 33% and 25% of patients, respectively [35]. The mean OS of patients with methylated versus unmethylated APC and RASSF1A was significantly longer (81 vs. 27 months and 71 vs. 46 months, respectively) in both metastatic and non-metastatic CRC.

A Danish study group selected 30 gene promotor regions for analysis and scored the number of hypermethylated regions [37]. A significant association was found between the presence of more than four methylated promotor regions and OS. In multivariate Cox regression, hypermethylation of RARB or RASSF1A was significantly associated with worse survival (HR 2.53, 95% CI 1.60–3.90). Liu et al. studied 7 different methylation markers, including SEPT9, in 165 patients [41]. A significant correlation was found between high methylated SST and risk of recurrence, DFS, and OS. No significant relationship with outcome was found for SEPT9. Recently, a diagnostic and prognostic score, including nine methylation markers, was created [47]. Patients with a high combined prognostic score had significantly worse OS, but over 50% of included patients had stage IV disease. In addition, an editorial expression of concern considered the uncertainty about methods of methylation cg10673833 analysis.

Instead of focusing on hypermethylation, Xue and colleagues studied hypomethylation of cystathionine-beta-synthase (CBS), which could be detected in 64% of patients [36]. Compared to high plasma methylation levels (HPM), low plasma methylation levels (LPM) were associated with higher recurrence risk (60.38% vs. 26.19%) and decreased OS (RR 1.35, 95% CI 1.09–2.41), both in univariate and multivariate analysis.

## 4. Discussion

To our knowledge, this is the first systematic review that focuses on the presurgery analysis of high levels of cfDNA or detection of ctDNA as a prognostic factor in non-metastatic CRC patients. A total of 29 articles including 3746 CRC patients met the criteria for this review and are categorized into three groups that reflect the type of approach: measurement of cfDNA, ctDNA somatic alterations, and ctDNA methylation. We did not observe a consistent (lack of) association between presurgery cfDNA/ctDNA results and outcomes. In a heterogeneous landscape of methods, assays, and study populations, both positive and negative associations are reported. Aggregation of study results is hampered by a lack of comparability between studies. Therefore, the question of the presurgery cfDNA/ctDNA analysis has a prognostic value remains yet unanswered.

Important assay variations exist between the various approaches to detect cfDNA/ctDNA, each approach having its own advantages and disadvantages. First of all, quantifying cfDNA is the least specific, as other processes such as inflammation also impact cfDNA levels. In non-metastatic CRC patients, the amount of ctDNA is usually low, comprising less than 1% of total cfDNA. Moreover, cfDNA levels are increased by inflammation or recent surgery, which makes cfDNA difficult to interpret. Therefore cfDNA analysis is less likely to be an adequate prognostic or predictive biomarker. The second approach, detecting somatic alterations in blood, is more challenging than quantifying cfDNA but highly specific for cancer. Theoretically, the potential to predict the outcome with the analysis of somatic alterations seems more prominent than with the quantification of cfDNA, but several studies did not find any association. As some mutations are known to be prognostic themselves, a positive correlation between ctDNA and outcome could also relate to these specific mutations that are included in most gene panels. To improve both sensitivity and specificity, many assays use a tumor-informed approach, which is a disadvantage in the pretreatment setting where only tissue biopsy material is available. Lastly, for methylation analysis, no tissue is needed to reach an accurate specificity. Nevertheless, it is unclear which methylation assay has the most powerful prognostic potential as many different CpG regions are being studied. The Epi ProColon detecting SEPT9 methylation has received FDA approval as a screening tool, but this review shows that its prognostic value is debatable. Currently, methylation assays in which thousands of methylation regions are targeted are developed for early cancer detection [59]. Further research is needed to study if these multiplex methylation assays are associated with prognosis.

An important limitation of this review is the impossibility of performing a meta-analysis due to these major differences in cfDNA/ctDNA analyses. Furthermore, the prognostic value compared with or in addition to known prognostic factors such as the TNM stage, should be investigated. Both cfDNA and ctDNA levels are higher in advanced stages, but only 41% of included studies reported a multivariate analysis to determine if cfDNA/ctDNA predicts outcome independently from the TNM stage. Moreover, most articles included not only stage I–III patients but also stage IV patients who were treated with curative intent. A previous meta-analysis showed that high cfDNA levels are associated with poor prognosis in mCRC patients, and event rates in this group are higher than in stage I–III patients. Therefore, mCRC patients should not be part of the study population to evaluate prognosis in the non-metastatic setting [52]. These conflicting and ambiguous results are in line with cfDNA/ctDNA studies in rectal cancer patients treated with neoadjuvant therapy and in other solid tumors such as lung and breast cancer [60,61,62]. Study results regarding cfDNA/ctDNA should not be extrapolated blindly from one tumor type to another as prognosis, treatment, and metastatic patterns differ between early-stage cancers. Moreover, ctDNA detection rates may vary significantly between tumor types [16,33].

In this systematic review, we excluded studies that focused on non-cfDNA components of liquid biopsy, such as CTCs and microRNAs. In the past years, reviews have been published that studied the prognostic value of these biomarkers. In 2017, Yang et al. performed a meta-analysis to analyze the prognostic significance of CTCs detected by RT-PCR in non-metastatic CRC patients [63]. CTC-positive status was associated with poor prognosis, regardless of sampling time (pre- or postsurgery). More recently, a correlation between CTC detection presurgery and prognosis was found in a German study [64]. The predictive value of CTCs, and therefore its clinical utility, has not been proven yet [65]. Circulating microRNAs have been less extensively studied in colorectal cancer. These studies suggest that microRNAs might be more useful as a diagnostic biomarker than as a prognostic biomarker [66,67]. The development of a presurgery cfDNA or ctDNA assay that predicts outcome could support the stratification of patients according to recurrence risk and would allow risk-stratified neoadjuvant treatment. Neoadjuvant strategies are currently being investigated for colon cancer patients in the FOxTROT and CONNECTION-II trials [68,69]. The benefits of neoadjuvant treatment in non-metastatic colon cancer are increasingly recognized and also supported by a recent meta-analysis [8]. This increases the clinical need for a pretreatment prognostic biomarker as the clinical TNM stage based on radiological imaging is too inaccurate [70].

From another point of view, the pretreatment cfDNA/ctDNA analysis could be of interest in addition to a postsurgery analysis to observe the dynamics of cfDNA/ctDNA. These dynamics could have a prognostic value in themselves. Data of serial ctDNA analyses in the postsurgery setting are available. For example, Wang et al. studied ctDNA dynamics during follow-up after surgical resection in stage I–III CRC patients. Three patients had detectable ctDNA levels after surgery that became undetectable during follow-up. These patients did not experience recurrence [71]. The Australian study of Tie et al. showed the prognostic value of detectable ctDNA after surgery and after adjuvant chemotherapy with a very poor prognosis in stage III colon cancer patients with persistently detectable ctDNA [72]. Future research is needed to study these dynamics, including a presurgery blood sample.

Analysis of cfDNA/ctDNA is considered an emerging biomarker in cancer care. However, in this systematic review, we show that, based on current literature, no major conclusions can be drawn about the potential of presurgery cfDNA/ctDNA to predict outcomes in colorectal cancer. Several steps need to be taken into account to answer this question. First of all, there is a need for studies that primarily focus on the question if pretreatment cfDNA/ctDNA is prognostic or predictive for outcomes and how this relates to known prognostic factors such as the TNM stage. Many ctDNA studies were excluded from this systematic review because presurgery samples were not collected or analyzed as the postsurgery blood sample was the focus of these studies. Secondly, a suitable quality comparison between the various cfDNA/ctDNA assays, including both mutation and methylation assays, is needed to determine the most favorable analysis strategy. Ideally, recent developments, including the aforementioned multiplex methylation assays but also cfDNA fragmentation profiles, are being included in these comparative studies [73].

We strongly encourage authors to follow available reporting guidelines to improve comparability and interpretation. The Enhancing the QUAlity and Transparency Of health Research (EQUATOR) network is an international initiative that aims to improve the reliability and value of published health research literature by promoting transparent and accurate reporting [74]. Reporting guidelines include the CONSORT guideline for randomized controlled trials and the STROBE statement for observational studies [75,76]. In addition, the REMARK checklist was developed to address widespread deficiencies in the reporting of tumor marker prognostic studies [77]. We believe these guidelines form a suitable basis for the reporting of cfDNA/ctDNA studies but, ideally, could be expanded into cfDNA/ctDNA-specific reporting guidelines. Ultimately, the clinical, pathological, and molecular data that are collected should be handled according to the FAIR (findable, accessible, interoperable, and reusable) data principles. These principles facilitate the reuse of research data [78].

## 5. Conclusions

We could not conclude whether the presence of high levels of cfDNA or detection of ctDNA before surgery is prognostic in non-metastatic CRC patients. Many different assays and biomarkers are being used, which hinders direct comparisons. Moreover, the additional value of cfDNA or ctDNA analysis to known prognostic risk factors as TNM stage remains unknown. There is a need for studies that primarily focus on whether pretreatment cfDNA/ctDNA is prognostic or predictive for outcomes. We also recommend reporting findings according to existing guidelines, such as the REMARK checklist. In addition, the FAIR data principles should be followed to promote optimal (re)use and comparability of research data. This will help to clarify the potential prognostic role of presurgery cfDNA/ctDNA, which could allow future improvements in the risk-stratified treatment of non-metastatic CRC.

## Figures and Tables

**Figure 1 cancers-14-02218-f001:**
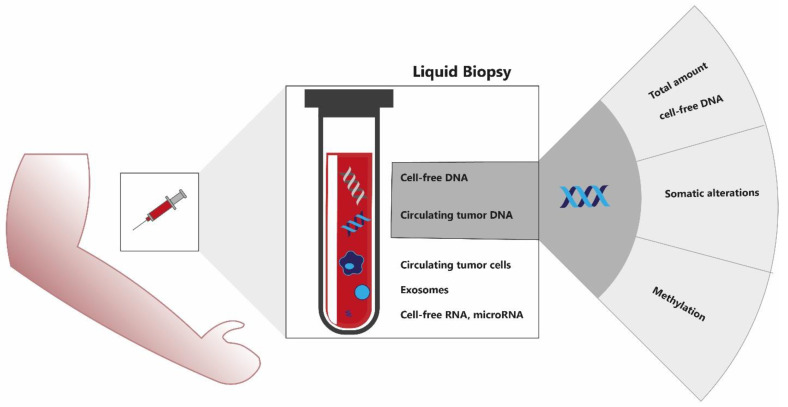
Liquid biopsy biomarker assays encompass a range of technologies to derive tumor information from body fluids, including cell-free DNA, circulating tumor DNA, circulating tumor cells, cell-free RNA, microRNAs, and exosomes. Several approaches are used to analyze circulating cell-free (tumor) DNA.

**Figure 2 cancers-14-02218-f002:**
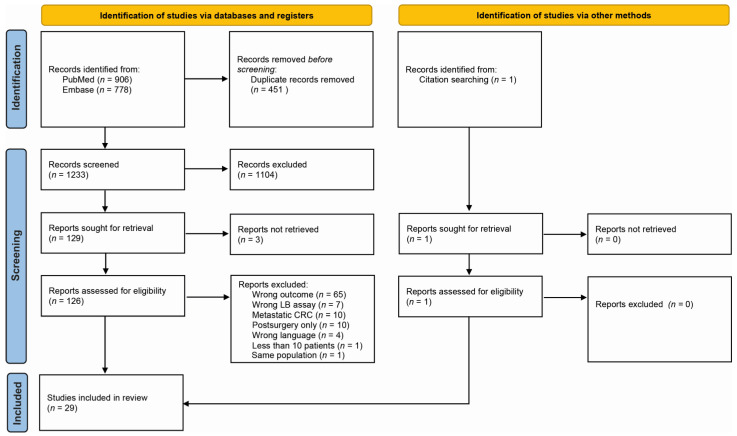
PRISMA flow chart of included studies in this systematic review.

**Table 1 cancers-14-02218-t001:** Summary of included studies on the association between circulating cell-free (tumor) DNA and prognosis.

Author, Year	Biomarkers	Assay	n	Tumor Stage	% Detectable Presurgery	Correlation with Stage	Outcome
Cell-Free DNA							
Czeiger et al., 2016 [20]	Cell-free DNA	SYBR Gold fluorometry	38	I: n = 5	100% detectable	Higher levels in stage IV	mHR for DFS = 6.03 (95% CI 1.87–19.41)
				II: n = 20	49% above cutoff of 800 ng/mL		mHR for OS = 3.53 (95% CI 1.46–8.55)
				III: n = 7			
				IV: n = 5			
Guadalajara et al., 2008 [21]	Cell-free DNA	Spectrophotometry (NanoDrop)	73	I: n = 17	Not reported.	Higher levels in stage IV	No significant correlation between cfDNA concentration and development of metastases or mortality. Trend toward worse prognosis for patients with cfDNA concentration >60 ng/μL
				II: n = 25			
				III: n = 19			
				IV: n = 11			
				Benign: n = 1			
Fleming et al., 2020 [22]	Cell-free DNA	Spectrophotometry (NanoDrop)	20	I–II: n = 9	Not reported	Not reported	Slightly higher cfDNA levels in patient with a recurrence compared to non-recurrence patients
				III: n = 11			
Zhong et al., 2020 [23]	Cell-free DNA	qPCR	60	I–II: n = 26	Not reported.	Yes	cfDNA concentration was an independent risk factor for PFS in both univariate and multivariate regression analysis
				III–IV: n = 34			
Somatic alterations							
Wang et al., 2004 [24]	APC, KRAS, TP53	PCR-SSCP tumor-naive, serum	104	I: n = 7 II: n = 49 III: n = 39 IV: n = 9	0.46	Non-significant trend	75% vs. 9.5% recurrences (*p* < 0.001)
Lecomte et al., 2002 [25]	KRAS (codon 12, 13) Also: cfDNA, p16 methylation	PCR tumor-informed, plasma	58	I: n = 8 II: n = 21 III: n = 16 IV: n = 13	cfDNA: 43% KRAS2: 45% p16: 68% Overall: 68% (stage I–III)	No	Significant worse RFS for ctDNA+ stage I–III patients: 2 y RFS of 66% (95% CI 36–84%) vs. 100%. mHR for OS in stage I–IV = 13 (95% CI 1.5–112).
Shin et al., 2017 [26]	KRAS	Sequenom MassARRAY + modified ultrahigh-sensitivity assay tumor-naive, plasma	160	I–II: n = 19 III: n = 35 IV: n = 106	17% in stage I–III	Correlation with heavier tumor burden	89% vs. 78% recurrences in stage I–III patients. Lower PFS (17 vs. 21 months), but not significant
Reinert et al., 2016 [27]	Patient-specific somatic structural variants	dPCR tumor-informed, plasma	11	I: n = 1 II: n = 5 III: n = 2 IV: n = 3	0.73	Non-significant trend	ctDNA+: 5/8 rec ctDNA−: 1/3 rec
Scholer et al., 2017 [28]	Patient-specific somatic structural variants and SNVs	dPCR tumor-informed, plasma	27	I: n = 5 II: n = 8 III: n = 8 IV: n = 6	0.74	Yes	8/10 ctDNA+ in stage I–III patients with relapse 6/11 ctDNA+ in stage I–III patients without relapse
Thomsen et al., 2017 [29]	RAS, BRAF	dPCR tumor-informed, serum	294	I: n = 40 II: n = 151 III: n = 103	0.42	Yes	RAS: mHR for DFS = 2.18 (95% CI 1.26–3.77). mHR for OS = 2.30 (95% CI 1.27–4.15). BRAF and pMMR: mHR for DFS = 3.61 (95% CI 1.70–7.67). mHR for OS = 3.45 (95% CI 1.52–7.85).
Tarazona et al., 2019 [30]	29 cancer-related genes	dPCR tumor-informed, plasma	94	I: n = 14 II: n = 41 III: n = 39	0.64	Lower levels in stage I	No relation between ctDNA and outcome: uHR for DFS = 0.93 (95% CI: 0.33–2.69)
Nakamura et al., 2021 [31]	KRAS (codon 12, 13)	dPCR tumor-naive, plasma	180	I–III: n = 154 IV: n = 26	33% (30% in stage I–III)	Non-significant trend	Increased recurrence risk for ctDNA+ patients (27% vs. 3%). mHR for RFS = 2.18 (95% CI 1.02–4.61)
Reinert et al., 2019 [12]	Patient-specific mutations	Multiplex PCR-based NGS tumor-informed, plasma	125	I: n = 5 II: n = 39 III: n = 81	0.89	Lower levels in stage I	No significant association between ctDNA and outcome
Allegretti et al., 2020 [32]	15 cancer-related genes	Targeted NGS + dPCR tumor-naive, plasma	39	I: n = 9 II: n = 14 III: n = 11 NR: n = 5	0.44	Weak, non-significant trend	3/10 recurrences in follow-up patients. 3 recurrences: 100% ctDNA+ before surgery. 7 non-recurrences: 4/7 ctDNA+ before surgery
Phallen et al., 2017 [33]	58 cancer-related genes	Targeted NGS tumor-naive, plasma	42	I: n = 8 II: n = 9 III: n = 10 IV: n = 15	0.83	Lower levels in stage I	uHR for PFS/OS = 1.13 (95% CI 1.03–1.24) in stage I–III mHR for PFS = 36.3 (95% CI 2.8–471.1) in stage I–IV
Suzuki et al., 2020 [34]	52 cancer-related genes	Targeted NGS tumor-naive, plasma	154	I: n = 29 II: n = 64 III: n = 50 IV: n = 11	0.73	Non-significant trend	4 recurrences in CRC patients with detectable ctDNA before surgery. MAF heat plot does not discriminate between recurrence and non-recurrence patients
Methylation							
Matthaios et al., 2016 [35]	APC, RASSF1A methylation	PCR	155	I–III: n = 88 IV: n = 67	APC: 33% RASSF1A: 25%	Yes	APC: OS 27 vs. 81 months RASSF1A: OS 46 vs. 71 months (*p* < 0.001)
Xue et al., 2017 [36]	Cystathionine-beta-synthase (CBS) hypomethylation	PCR	95	I: n = 10 II: n = 22 III: n = 38 IV: n = 15	0.64	Yes	RR of RFP = 1.54 (95% CI 1.18–3.02) RR of OS = 1.35 (95% CI 1.09–2.41)
Rasmussen et al., 2018 [37]	30 gene promotor regions	PCR	193	I–III: n = 159 IV: n = 34	NR	Non-significant trend	Signification association between OS and >4 methylated regions RARB or RASSF1A: mHR for OS = 2.53 (95% CI 1.60–3.90)
Lin et al., 2015 [38]	TWIST1, FLI1, AGBL4	qPCR (Sequenom MassArray)	353	I: n = 42 II: n = 140 III: n = 108 IV: n = 63	≥1 meth: 93% AGBL4: 65% FLI1: 66% TWIST1: 70%	No	No significant association between (number of) methylated genes and DFS
Wallner et al., 2006 [39]	TMEFF2, HLTF, hMLH1	qPCR	77	I: n = 10 II: n = 24 III: n = 24 IV: n = 15	HLTF: 22% HPP1: 12% hMLH1: 23%	Yes	TMEFF2 or HLTF: mRRD = 3.4 (95% CI 1.4–8.1)
Herbst et al., 2009 [40]	HLTF, TMEFF2	qPCR	106	I: n = 13 II: n = 39 III: n = 54	HLTF: 12% TMEFF2: 6%	No	HLTF: mRRR = 2.5 (95% CI 1.1–5.6). Significant worse RFS (*p* = 0.014).
Liu et al., 2016 [41]	SST, MAL, TAC1, SEPT9, EYA4, CRABP1, NELL1	qPCR	165	I: n = 26 II: n = 62 III: n = 62 IV: n = 15	0.5	NR	mSST: mHR for DFS = 2.60 (95% CI 1.37–4.94) mSST: mHR for CSD = 1.96 (95% CI 1.06–3.62)
Bedin et al., 2017 [42]	SFRP1, OSMR Also: total amount cfDNA	qPCR	114	I: n = 38 II: n = 29 III: n = 32 IV: n = 15	0.67	Methylation: No cfDNA: Higher levels in stage III/IV	Methylation: no significant association with DFS/OS cfDNA: high level associated with poor prognosis. mHR for OS ALU83 = HR 2.71 (95%CI 1.22–6.02), mHR for OS ALU244 = 2.40 (95% CI 1.11–5.19).
Song et al., 2018 [43]	SEPT9	qPCR	120	I: n = 14 II: n = 40 III: n = 45 IV: n = 21	0.87	Yes	uHR for OS = HR 2.51 (95% CI 1.03–6.12)
Constâncio et al., 2019 [44]	APC, FOXA1, GSTP1, HOXD3, RARβ2, RASSF1A, SEPT9, SOX17	qPCR	100	I–II: n = 39 III: n = 43 IV: n = 18	SEPT9: 8% SOX17: 11%	Higher levels in stage IV	Significant association between RARβ2, SEPT9 and SOX17 and DSM
Leon Arellano et al., 2020 [45]	SEPT9	qPCR	10	II: n = 4 III: n = 3 IV: n = 3	0.8	NR	No significant association with recurrence ctDNA+: 1/5 recurrence ctDNA−: 0/2 recurrence
Jin et al., 2021 [46]	SEPT9	qPCR	82	I: n = 5 II: n = 30 III: n = 40 IV: n = 7	0.89	Higher levels for stage III and IV	No significant association with recurrence
Luo et al., 2020 [47]	Diagnostic score (cd-score) including 9 methylation markers	Targeted NGS + dPCR	801	I: n = 38 II: n = 139 III: n = 209 IV: n = 406	0.88	Higher levels for stage III and IV	mHR for OS = 2.24 (SE 0.11)

PCR: polymerase chain reaction. dPCR: digital PCR. ddPCR: digital droplet PCR. NGS: next-generation sequencing. qPCR: quantitative PCR. MASA: mutant allele-specific amplification. SSCP: single-strand conformation polymorphism. mHR: hazard ratio in multivariate analysis. uHR: hazard ratio in univariate analysis. DFS: disease-free survival. OS: overall survival. PFS: progression-free survival. RFS: recurrence-free survival. RR: relative risk. mRRD: RR of death in multivariate analysis. RFP: recurrence-free probability. MAF: mutant allele fraction. pMMR: proficient mismatch repair. DSM: disease-specific mortality. CSD: cancer-specific death.

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
