# Peer review of "Cell-Free Circulating (Tumor) DNA before Surgery as a Prognostic Factor in Non-Metastatic Colorectal Cancer: A Systematic Review"

_cancers, 2022, doi:10.3390/cancers14092218_

Round 1

Reviewer 1 Report

In this manuscript, the authors conducted systemic review for the potential of cfDNA/ctDNA analysis before surgery to predict risk of recurrence in non-metastatic colorectal cancer patients. Although clear association was not concluded from the included 29 studies, this review is meaningful by focusing on the liquid biopsy at the timing of before surgical resection, which would be clinically valuable in the decision of neoadjuvant therapies. To improve this manuscript, below I suggest a few points.

Although this review excluded “circulating tumor cells”, it might indicate shedding of cancer cells into blood circulation and high possibility of micro-metastasis in other organs that can be related with poor prognosis. How’s about adding short discussion for the circulating tumor cells?

In the category of ctDNA analysis for somatic alterations, I wonder if a certain of genetic alterations or mutation burden can be associated with the prognosis.

There may remain one question whether the prognostic potential of pre-surgery cfDNA/ctDNA analysis can be beneficial in another type of cancer. Conclusion of this review would be a consensus idea regardless of cancer type?

Author Response

Dear reviewer,

We thank you for the careful evaluation of our manuscript. We adjusted the manuscript according to the valuable comments and addressed the issues that were raised point by point:

Point 1: Although this review excluded “circulating tumor cells”, it might indicate shedding of cancer cells into blood circulation and high possibility of micro-metastasis in other organs that can be related with poor prognosis. How’s about adding short discussion for the circulating tumor cells?

Response 1: We deliberately decided to focus on cell free DNA, including circulating tumor DNA and cfDNA methylation markers as plasma biomarkers in colorectal cancer and excluded circulating tumour cells from our search. This way, we tried to demarcate and limit our search in a continously evolving field. However, we agree it has additional value to explicitly mention CTCs in our discussion as another relevant liquid biopsy biomarker with potential clinical applicability, because presence of CTCs in blood is associated with an increased risk of recurrence and poor prognosis in CRC, similar to our plasma-based biomarkers. Therefore we added a paragraph in which the current knowledge about CTCs is taken into account (Discussion, fourth paragraph – page 14/15, line 393-400):

In this systematic review we excluded studies that focused on non-cfDNA components of liquid biopsy, such as CTCs and microRNAs. In the past years, reviews have been published that studied the prognostic value of these biomarkers. In 2017, Yang et al. performed a meta-analysis to analyze the prognostic significance of CTCs detected by RT-PCR in non-metastatic CRC patients. CTC-positive status was associated with poor prognosis, regardless of sampling time (pre- or postsurgery). More recently, a correlation of CTC detection presurgery and prognosis was found in a German study. The predictive value of CTCs, and therefore its clinical utility, has not been proven yet. Circulating microRNAs have been less extensively studied in colorectal cancer, but few studies suggest that microRNAs might be more useful as a diagnostic biomarker than as a prognostic biomarker.”

Point 2: In the category of ctDNA analysis for somatic alterations, I wonder if a certain of genetic alterations or mutation burden can be associated with the prognosis.

Response 2: It is known that presence of certain mutations, for example BRAFV600E, is associated with a poor prognosis. In the context of liquid biopsies in non-metastatic disease, the question is whether the outcome depends on the specific mutation status or on the fact that circulating tumor DNA is detectable regardless of the type of mutations found. We tried to emphasize this in more detail in our discussion (Discussion, second paragraph – page 14, line 364-366):

As some mutations are known to be prognostic itself, a positive correlation between ctDNA and outcome could also relate to these specific mutations that are included in most gene panels.”

Point 3: There may remain one question whether the prognostic potential of pre-surgery cfDNA/ctDNA analysis can be beneficial in another type of cancer. Conclusion of this review would be a consensus idea regardless of cancer type?

Response 3: We hypothesize that the potential of presurgery cfDNA/ctDNA analysis is also valid in other types of cancer. In our discussion we refer to comparable studies in breast and lung cancer, in which presurgery samples are being investigated. Also in these tumor types the liquid biopsy studies in the non-metastatic setting are heterogeneous and conclusions cannot be drawn yet. However, we believe that the applicability of cfDNA/ctDNA should be established for each tumor type independently for several reasons. First of all, different cancer types have different prognosis and treatment in the non-metastatic setting. Second, it must be taken into account that different tumor types possibly differ in the extent of measurable cfDNA/ctDNA in the non-metastatic and metastatic setting (Bettegowda, Sci Transl Med 2014). In fact, even within colorectal cancer differences in ctDNA detection are found between metastatic sites (Van’t Erve, J Pathol Clin Res 2021). We added these considerations to our discussion (Discussion, third paragraph – page 14, line 364-367):

“Study results regarding cfDNA/ctDNA should not be extrapolated blindly from one tumor type to another as prognosis, treatment and metastatic patterns differ between early stage cancers. Moreover, ctDNA detection rates may vary significantly between tumor types.”

Kind regards,

Suzanna Schraa, MD

Reviewer 2 Report

Revision of systematic review entitled “Cell-Free Circulating (Tumor) DNA before Surgery as a Prog-2 nostic Factor in Non-Metastatic Colorectal Cancer: A Systematic 3 Review” by Suzanna J. Schraa et al.

This systematic review aims to provide an overview of publications in which the prognostic value of presurgery cfDNA/ctDNA in non-metastatic CRC patients was studied and is performed according to PRISMA guidelines. 29 out of 1233 articles were included and categorized in three 34 groups that reflect the type of approach: measurement of cfDNA, ctDNA somatic alterations and ctDNA methylation. Overall, a clear association between presurgery cfDNA/ctDNA and outcome was not observed, but large studies that primarily focus on the prognostic value of presurgery cfDNA/ctDNA are lacking. The authors concluded that standardized studies on the value of presurgery cfDNA/ctDNA are needed to improve comparability and interpretation among studies.

Although the review is clearly writing, is well conducted and structured, the final number of the article analyzed is very small, especially in some subcategories. It should be interesting complete the analysis adding two other categories, the circulating tumor cells and microRNAs. Are these two categories used as a prognostic pre-surgery marker for metastatic colorectal-cancer?  

Author Response

Dear reviewer,

We thank you for the careful evaluation of our manuscript. We adjusted the manuscript according to the valuable comments and addressed the issues that were raised point by point:

Point 1: Although the review is clearly writing, is well conducted and structured, the final number of the article analyzed is very small, especially in some subcategories. It should be interesting complete the analysis adding two other categories, the circulating tumor cells and microRNAs. Are these two categories used as a prognostic pre-surgery marker for metastatic colorectal-cancer? 

Response 1:

Thank you for your suggestion. We excluded both circulating tumor cells and microRNA from our search to not further extend the heterogeneity of the papers included. We focused on circulating cell free DNA because this seems to be one of the most promising biomarker to be implemented in clinical practice within the coming years. We consider circulating tumor cells and microRNAs as different entities within liquid biopsy research with its own technical and clinical challenges. To our knowledge, these categories are not being used as a prognostic marker in metastastic colorectal cancer at this moment. We do agree it has additional value to discuss circulating tumor cells and microRNA as they are relevant and promising biomarkers in blood. Therefore we added a paragraph to our discussion (Discussion, fourth paragraph – page 14/15, line 393-403).

In this systematic review we excluded studies that focused on non-cfDNA components of liquid biopsy, such as CTCs and microRNAs. In the past years, reviews have been published that studied the prognostic value of these biomarkers. In 2017, Yang et al. performed a meta-analysis to analyze the prognostic significance of CTCs detected by RT-PCR in non-metastatic CRC patients. CTC-positive status was associated with poor prognosis, regardless of sampling time (pre- or postsurgery). More recently, a correlation of CTC detection presurgery and prognosis was found in a German study. The predictive value of CTCs, and therefore its clinical utility, has not been proven yet. Circulating microRNAs have been less extensively studied in colorectal cancer, but few studies suggest that microRNAs might be more useful as a diagnostic biomarker than as a prognostic biomarker.”

Kind regards ,

Suzanna Schraa, MD

Reviewer 3 Report

This systematic review is well-conducted, written and presented. A few typos or minor errors will need to be corrected. 

Author Response

Dear reviewer,

We thank you for the careful evaluation of our manuscript. We adjusted the manuscript according to the valuable comments and addressed the issues that were raised point by point:

Point 1: This systematic review is well-conducted, written and presented. A few typos or minor errors will need to be corrected.

Response 1:

We reviewed our manuscript and corrected spelling and grammar mistakes.

Kind regards ,

Suzanna Schraa, MD